# Research on the Dynamic Characteristics of a Dual Linear-Motor Differential-Drive Micro-Feed Servo System

Hanwen Yu *, Guiyuan Zheng, Yandong Liu *, Jiajia Zhao, Guozhao Wei and Hongkui Jiang

School of Mechanical and Electronic Engineering, Shandong Jianzhu University, Jinan 250101, China; zgy18764977064@163.com (G.Z.); zhaojia0922@126.com (J.Z.); wgz_jn@sdjzu.edu.cn (G.W.); jhk_2001@163.com (H.J.)
* Correspondence: yuhanwen20@sdjzu.edu.cn (H.Y.); yandonliu@foxmail.com (Y.L.)

**Featured Application: The potential applications of the dual linear-motor differential-drive system and the numerical analysis results are applicable to high-end equipment fields like suspensions, robotics, optoelectronics, powertrain, integrated electronics, national defense, and genetic engineering, enabling the new system to have a lower stable micro-feed velocity and attain accurate micro-feed control.**

**Abstract:** (1) Objectives: This article presents a dual linear-motor differential drive micro-feed servo system, mainly through the optimization design of the transmission mechanism. Owing to the differential synthesis of the micro feed from the upper and under linear motors, the impact of friction nonlinearity during the ultra-low velocity micro feed is avoided, endowing the system with a lower stable feed speed to achieve precise micro-feed control. (2) Methods: Transmission components of the dual linear-motor differential-drive system are analyzed using the lumped parameter method, and a dynamic model of electromechanical coupling is created, which takes into account nonlinear friction. The motion relationship of the dual linear-motor differential-drive servo feed system is characterized using a transfer function block diagram. (3) Discussions: Through simulation, the differences in response between the linear-motor single-drive system and the dual linear-motor differential-drive system are examined under fixed or variable feeding velocities as well as the impact of varying velocity combinations of dual linear motors on the output speed of the differential drive system. (4) Results: Nonlinear friction factors exert an impact on the feed velocity of both linear-motor single-drive and dual linear-motor differential-drive systems during low-velocity micro feed. However, regardless of the constant or variable speed conditions, the dual linear-motor differential-drive servo system significantly outperforms the linear-motor single-drive system regarding low-velocity micro feed. Our simulation results are basically consistent with engineering practice, thus validating the rationality of the created system models, which paves the ground for the micro-feed control algorithms.

**Keywords:** differential dual drive; linear motor; micro-feed system; nonlinear friction; dynamic characteristics

## 1. Introduction

One of the key technical bottlenecks in achieving ultra-precision machining is how to achieve accurate, stable, and reliable micro displacement of the tool or workpiece during the machining process [1]. For most precision and ultra-precision machining machines, a high-performance linear motion system is necessary and critical [2]. The linear motor-drive system does not require any intermediate mechanical transmission mechanism, and the linear motor directly provides thrust to the table, eliminating the consumption caused by the transmission mechanism and the limitation of transmission clearance, achieving "zero transmission" from the motor to the table [3]. Moreover, it has the characteristics of fast response speed, small electrical time constant, high thrust, and low loss, which

can provide high dynamic response speed and acceleration as well as extremely high stiffness, high positioning accuracy, smooth and error-free motion, but cannot eliminate nonlinear friction [4]. Therefore, the low-speed linear motion of the table relative to the guide rail has become the main factor restricting the improvement of feed accuracy in CNC machine tools.

Based on the differential motion synthesis mechanism of dual linear motors and modern servo drive control technology, a micro-feed servo system with dual linear-motor differential drives is proposed. By superimposing two quasi-equal macroscopic movements, namely "the upper linear motor drives the upper table to move in a straight line" and "the under linear motor drives the under table to move in a straight line", which have almost equal instantaneous velocities and opposite directions, the inevitable low-speed nonlinear crawling phenomenon caused by the inherent properties of the traditional electromechanical servo system structure is avoided, and high-precision differential micro-feed control is achieved, fundamentally eliminate the creeping of linear motion in micro-feed mechanisms.

Many scholars use the lumped parameter method to model electromechanical transmission systems [5–12]. Jin, HY [5] established the mathematical model of a permanent magnet linear synchronous motor (PMLSM) with uncertainties, that is, a second-order complementary sliding mode control (SOCSMC) with fast convergence and global robustness, to conquer the uncertainties and reduce chattering. Golzarzadeh M [6] presented a comprehensive thermal model based on the lumped parameter approach for STLSRM, which predicts temperature distribution in different parts of this motor, including slot winding, end-winding, stator pole, stator yoke, and the moving part. Ullah W [7] aims to review analytical methodologies, i.e., lumped parameter magnetic equivalent circuit (LPMEC), magnetic co-energy (MCE), Laplace equations (LE), Maxwell stress tensor (MST) method, and sub-domain modeling for the design of a segmented PM (SPM) consequent pole flux switching machine (SPMCPFSM). Far MF [8] proposes a fast-dynamic model, based on a model order reduction method, to control a permanent magnet synchronous machine at a wide range of speeds. The robustness of the control is observed particularly when the linear lumped parameter-based models are employed to represent a machine composed of nonlinear magnetic materials. Ullah N [9] combined the merits of PMLFSM and FELFSM by proposing a novel Hybrid Excited LFSM (HELFSM), the proposed machine is excited by PMs, Field Excitation Coils (FECs), and Armature Windings (AWs). Waheed A [10] designed a two-pole, three-phase, 7.5-kW line-start permanent magnet (LSPM) synchronous motor, which is an analytical technique for rotor geometry optimization based on a lumped magnetic parametric approach. For the palletizing robot's operating characteristics of high speed, high acceleration, and heavy load, Yu HW [11] carried out kinematics analysis by using the lumped parameter method, which obtained a positive kinematics solution and workspace. Zhu Y [12] proposed an accurate and simple five-node lumped parameter thermal network (LPTN) and built the mathematical model of the LPTN, aiming at resolving the difficulty in online temperature estimation. However, the above research results are all aimed at linear feed single-drive systems, and the dynamic model of a dual linear-motor differential-drive system is not established.

Many research achievements have been made on the influence of nonlinear friction on micro-feed systems [13–18]. Luna L [13] introduced a delay-based nonlinear controller aimed at position control of linear ultrasonic motors, which is termed the cascade nonlinear proportional integral retarded control law, to achieve high precision and fast response in spite of the effects of inner disturbances, hysteresis, and friction phenomena. Zhang W [14] not only incorporated various non-linear factors such as time-varying friction forces, time-varying mesh stiffness, and damping associated with internal excitations but also placed significant emphasis on analyzing the impact of non-linear external excitations, including road surface unevenness and motor torque fluctuations, on the shifting process of the drive system. The friction force at the contact interface is easily affected by factors such as surface morphology, friction coefficient, and sliding speed, especially when the linear motor is running at low speed, Li H [15] proposed an orthogonal vibration decoupling

method for low-speed regulation. Li YR [16] presented an asymmetric friction model and an indirect integral method (IIM), the asymmetric friction model is able to capture the nonlinear position drifting phenomenon. Yang QY [17] proposed a novel design method for permanent magnet linear motor systems based on backstepping, in order to solve the influence of unknown factors on the system parameters of a permanent magnet linear motor including nonlinear friction, sudden load changes, thrust fluctuations, and so on. In order to avoid discontinuities, the nonlinear friction of the motor, which includes static, coulomb, and viscous terms, is considered a smooth function. Lee KH [18] presented a state observer for an elastic joint with nonlinear friction via the information from an acceleration sensor.

Many papers have established physical and mathematical models for dual servo systems [19–25]. Chang H [19] proposed a topological structure expanded by five improved sliding-mode observers (ISMOs) to simultaneously identify the load speed, the dual motor inertia, the load inertia, the stiffness coefficient, and the load disturbance. Chen YZ [20] proposed a novel two degrees of freedom, large range, coarse-fine parallel dual-actuation flexure micropositioner (CFPDFM) with low interference behavior. Shang DY [21] investigated a difficult problem of nonlinear dynamics and motion control of a dual-flexible servo system with an underactuated hand (DFSS-UH) and designed a novel neural network sliding mode control (NNSMC) method to control the DFSS-UH. Yang XY [22] proposed a finite-time tracking and synchronization control method for dual-motor servo systems that suffer from backlash and time-varying uncertainties, to overcome several factors that may degrade the system's performance, such as transmission backlash, parameter drift, and motor dynamic characteristic differences. Wang BF [23] investigated the finite-time command-filtered backstepping control problem for dual-motor servo systems. The advantages of the finite-time controller include fast convergence and high robustness, which can improve the dynamic and steady control performance of the system. Yu HW [24] presented the design for a new differential dual-drive low-speed micro-feed mechanism and studied the difference in response of single-drive and differential dual-drive systems under the influence of friction and clearance. Jiang H [25] proposed a useful method for eliminating the gear clearance of the C axis of heavy-duty machine tools based on a dual servo-motor driving system, and established the dynamic model of the driving system of the dual servo motor, so as to find the non-linearity of the clearance, wear, and tooth clearance in the drive system. The above papers lack research on the transmission performance of dual linear-motor differential-drive micro-feed servo systems.

This paper establishes physical and mathematical models of the micro-feed servo system driven by linear motors and considers the influence of nonlinear friction on the system. The model comprehensively considers the servo stiffness of the linear motor and the contact stiffness between the linear motor actuator and the table. The influence of nonlinear friction on the dynamic characteristics of a dual linear-motor differential-drive system and a linear-motor single-drive system under constant speed and variable speed operating conditions are analyzed through numerical simulation.

## 2. Configuration of the Dual Linear-Motor Differential-Drive System

Figure 1 shows the configuration of the dual linear-motor differential-drive system [26], components as follows: 1—Fixed end of upper linear-motor drag chain, 2—Upper linear-motor actuator, 3—Moving end of upper linear-motor drag chain, 4—Upper linear-motor drag chain, 5—Upper linear-motor table, 6—Linear guide rail, 7—Proximity switch, 8—Mechanical limit block, 9—Upper linear-motor end cover, 10—Upper linear-motor stator, 11—Upper linear-motor assembly, 12—Under linear-motor stator, 13—Under linear-motor actuator.

This paper provides a dual linear-motor differential-drive micro-feed servo system. Due to the two high-speed "macro movements" that operate above the critical creeping speed, the "micro movements" are obtained through the differential speed of the upper and under linear motors. Therefore, it can eliminate the unavoidable and low-speed nonlinear creeping phenomenon caused by the inherent properties of the traditional electromechani-

cal servo-system structure, and enable the system to have a lower stable speed limit and achieve accurate micro-feed control. The upper linear motor drives the upper table to perform linear motion, while the under linear motor drives the under table to perform linear motion. According to the given motion requirements, specific algorithms are assigned to the upper linear motor and under linear motor motion commands to control the motion of the upper and under table, respectively.

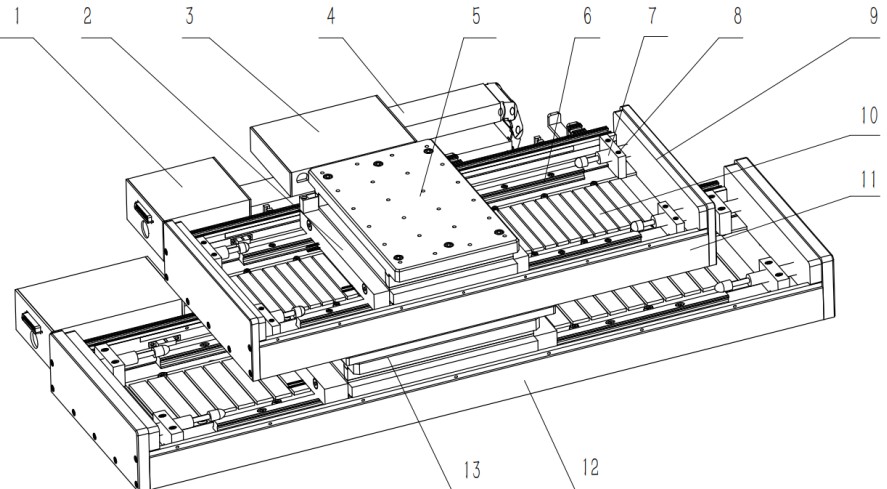

**Figure 1.** Dual linear-motor differential-drive precision transmission mechanism: 1—Fixed end of upper linear-motor drag chain, 2—Upper linear-motor actuator, 3—Moving end of upper linear-motor drag chain, 4—Upper linear-motor drag chain, 5—Upper linear-motor table, 6—Linear guide rail, 7—Proximity switch, 8—Mechanical limit block, 9—Upper linear-motor end cover, 10—Upper linear-motor stator, 11—Upper linear-motor assembly, 12—Under linear-motor stator, 13—Under linear-motor actuator.

The linear motion speed of the upper table driven by the upper linear motor alone along the axial direction is represented by V1, while the linear motion speed of the under table driven by the under linear motor alone along the axial direction is represented by V2. Under the drive of the dual linear motors, the differential composite speed of the two tables is close to zero, that is $\Delta = V1 - V2 \approx 0$, which avoids the creeping phenomenon caused by the low-speed movement of the table when driven by a single linear motor, allowing the dual linear-motor differential-drive system to achieve a high-precision micro-feed motion that a linear-motor single-drive servo system cannot achieve.

## 3. Dynamic Model of the Dual Linear-Motor Differential-Drive System

### 3.1. Mechanical Model

The dual linear-motor differential-drive micro-feed servo system exploits the motion-synthesis principle along with the servo-drive technique. It is obtained with the utilization of upper and under linear-motor single-drive subsystems, thereby achieving differential speed in the identical direction. Hence, establishing mathematical models separately for the linear-motor single-drive system and the dual linear-motor differential-drive system is required. For our dual linear-motor differential-drive micro-feed servo system, its transmission mechanism (displayed in Figure 1) is presented as the mechanical model of a flexible structure system created using the lumped parameter approach as depicted in Figures 2 and 3.

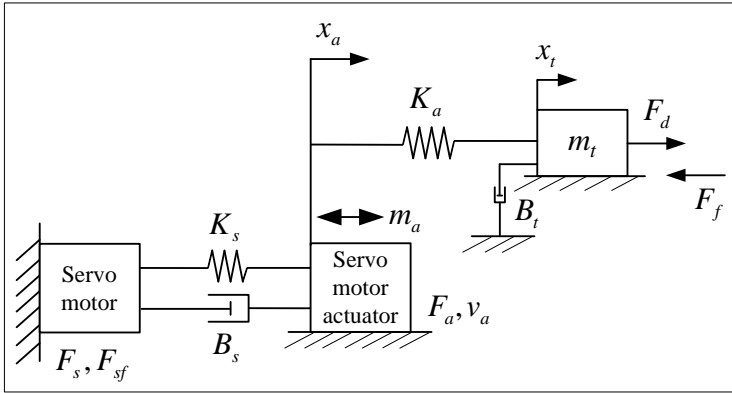

**Figure 2.** Mechanical model of linear-motor single-drive system.

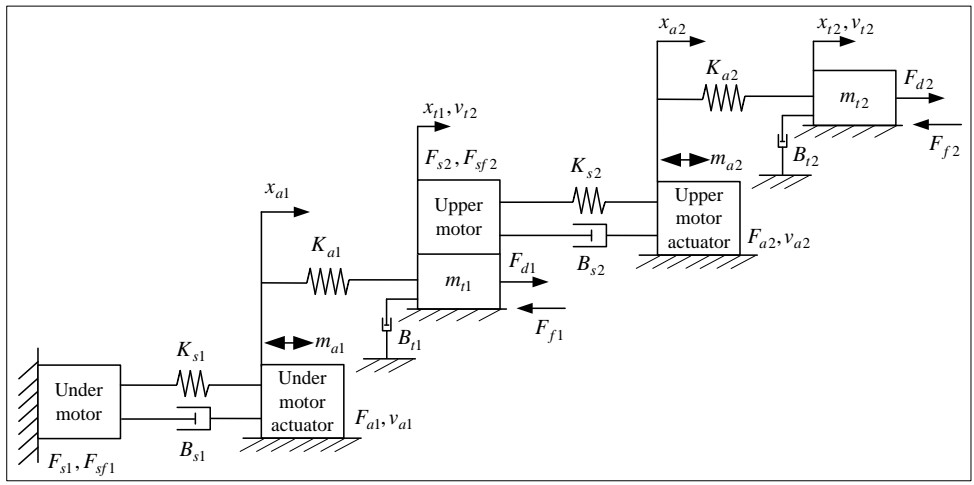

**Figure 3.** Mechanical model of dual linear-motor differential-drive system.

### 3.1.1. Linear-Motor Single-Drive System Analysis

The linear motor actuator is connected to the table, and the stator is fixed on the machine frame. The actuator generates electromagnetic thrust under the action of the current, which drives the connected table to achieve linear motion in the feed direction along the guide rail. As shown in Figure 2, the mechanical model of the linear-motor single-drive feed servo system is presented.

$$F_s = m_a \ddot{x}_a + B_s \dot{x}_a + F_{sf} + F_a \tag{1}$$

$$F_a = \frac{F_d}{\eta} \tag{2}$$

$$F_d = m_t \ddot{x}_t + B_t \dot{x}_t + F_f \tag{3}$$

$$x_t = x_a - \frac{F_d}{K_{eq}} \tag{4}$$

$$K_{eq} = \frac{K_s K_a}{K_s + K_a} \tag{5}$$

where $F_s$ is the servo driving force generated by the linear motor. $F_{sf}$ is the electromagnetic resistance generated on the linear motor. $F_a$ is the driving force exerted by the linear motor on the actuator, which generates the driving force $F_d$ acting on the table. $F_f$ is the equivalent coulomb friction force acting on the table when the linear-motor single drive is in use. $m_a$ represents the linear motor actuator's equivalent mass. $m_t$ represents the table's equivalent mass. $K_s$ represents the servo stiffness acting on the linear motor actuator.

$K_a$ represents the contact stiffness between the linear motor actuator and the table. $K_{eq}$ represents the comprehensive equivalent stiffness of the linear-motor single-drive system. $B_s$ represents the equivalent viscous friction coefficient acting on the linear motor actuator. $B_t$ represents the viscous friction coefficient between the guide rail and slider when the linear-motor single drive is in use. $x_a$ represents the theoretical displacement generated by the linear motor actuator. $x_t$ represents the actual displacement generated by the table when the linear-motor single drive is in use. $\eta$ is the mechanical efficiency of the linear-motor single-drive system.

### 3.1.2. Dual Linear-Motor Differential-Drive System Analysis

As shown in Figure 3, the mechanical model of the dual linear-motor differential-drive servo-feed system is presented. In order to ensure that the transmission parameters of the two single-drive systems are consistent, two linear motors with identical parameters are used in the design, and it is ensured that the equivalent mass and viscous friction coefficient of the upper linear motor actuator during the use of the upper linear-motor single drive are equal to the equivalent mass and viscous friction coefficient of the under linear motor actuator during the use of the under linear motor single drive, respectively, namely, $m_a = m_{a1} = m_{a2}$, $B_s = B_{s1} = B_{s2}$. At the same time, it is ensured that the equivalent mass and viscous friction coefficient of the upper table during the use of the upper linear-motor single drive are equal to the equivalent mass and viscous friction coefficient of the under table during the use of the under linear-motor single drive, respectively, that is $m_t = m_{t1} = m_{t2}$, $B_t = B_{t1} = B_{t2}$. In addition, it is required that the comprehensive transmission stiffness of the two single-drive systems and the differential-drive system be consistent, that is $K_{eq} = K_{eq1} = K_{eq2}$.

The dual linear-motor differential-drive system is achieved by comparing the difference values in force, speed, and displacement between the upper linear-motor single drive and the under linear-motor single drive, respectively.

$$\Delta F_s = F_{s1} - F_{s2} \tag{6}$$

$$\Delta F_a = F_{a1} - F_{a2} \tag{7}$$

$$\Delta F_d = F_{d1} - F_{d2} \tag{8}$$

$$\Delta F_{sf} = F_{sf1} - F_{sf2} \tag{9}$$

$$\Delta F_f = F_{f1} - F_{f2} \tag{10}$$

$$\Delta v_t = v_{t1} - v_{t2} \tag{11}$$

$$\Delta x_a = x_{a1} - x_{a2} \tag{12}$$

$$\Delta x_t = x_{t1} - x_{t2} \tag{13}$$

where $F_{s1}, F_{s2}$ are the servo driving forces generated by the upper linear motor and under linear motor, respectively. $F_{a1}, F_{a2}$ are the driving forces exerted by the upper linear motor and under linear motor on the actuator, respectively, and generate driving forces $F_{d1}, F_{d2}$ acting on the upper table and the under table, respectively. $F_{sf1}, F_{sf2}$ are the electromagnetic resistance generated on the upper linear motor and under linear motor, respectively. $F_{f1}, F_{f2}$ are the equivalent coulomb friction forces acting on the table when the upper linear motor and under linear motor are driven separately. $v_{t1}, v_{t2}$ respectively represent the actual speed of the table when the upper linear motor and under linear motor are driven individually. $x_{a1}, x_{a2}$ represent the theoretical displacement generated by the upper linear motor actuator and under linear motor actuator, respectively. $x_{t1}, x_{t2}$ respectively represent the actual displacement generated by the table when the upper linear motor and under linear motor are driven individually.

The equations of motion for the dual linear-motor differential-drive servo-feed system are Equations (14)–(18), then

$$\Delta F_s = m_a \Delta \ddot{x}_a + B_s \Delta \dot{x}_a + \Delta F_{sf} + \Delta F_a \tag{14}$$

$$\Delta F_a = \frac{\Delta F_d}{\eta} \tag{15}$$

$$\Delta F_d = m_t \Delta \ddot{x}_t + B_t \Delta \dot{x}_t + \Delta F_f \tag{16}$$

$$\Delta x_t = \Delta x_a - \frac{\Delta F_d}{K_{eq}} \tag{17}$$

$$K_{eq} = \frac{K_{s1} K_{a1}}{K_{s1} + K_{a1}} = \frac{K_{s2} K_{a2}}{K_{s2} + K_{a2}} \tag{18}$$

where $K_{s1}, K_{s2}$ represent the servo stiffness acting on the upper linear motor actuator and under linear motor actuator, respectively. $K_{a1}, K_{a2}$ respectively represent the contact stiffness between the upper linear motor actuator and the upper table, and the contact stiffness between the under linear motor actuator and the under table.

*3.2. Friction Model*

To create the table-guideway friction model, the friction on the table is solved with the utilization of Canudas' LuGre model [27,28], which enables the precise description of dynamic and static properties of diverse frictions, including presliding displacement, Stribeck and creeping effects, friction hysteresis as well as static friction alteration. As is displayed in Figure 4, the maximum static frictional force to sliding friction exhibits an ongoing alteration of negative damping characteristics.

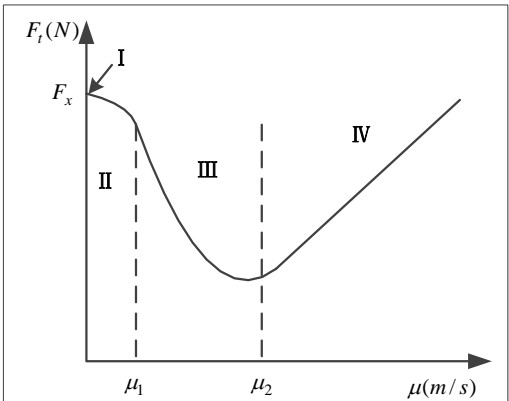

**Figure 4.** Stribeck curve [28]. I means the maximum static friction force, II means the solid/boundary friction region, III means the mixed friction region, and IV means the fluid friction region.

Let state quantity denote the average deformation quantity of the bristle contact surface, the friction on the table is formulated as follows:

$$F_f = \sigma_0 z + \sigma_1 \dot{z} + B_v v \tag{19}$$

$$\dot{z} = v - \frac{\sigma_0 |v|}{g(v)} z \tag{20}$$

$$g(v) = f_c + (f_s - f_c) \exp[-(\frac{v}{v_s})^2] \tag{21}$$

where $v$ and $v_s$ separately signify the relative and Stribeck velocities; $\sigma_0$, $\sigma_1$ and $B_v$ separately denote the rigidity, damping, and viscous friction coefficients; $f_c$ and $f_s$ separately represent

the coulomb and maximum static frictions. The function $g(v)$, which depends on multiple factors like lubrication, temperature, and material properties, exceeds zero.

When the system state is steady, then $\dot{z} = 0$, the correlation of friction force $F_f$ with relative velocity $v$ can be formulated as follows:

$$F_f = \left\{ f_c + (f_s - f_c) \exp\left[-\left(\frac{v}{v_s}\right)^2\right] \right\} \mathrm{sgn}(v) + B_v v \tag{22}$$

### 3.3. AC Servo Motor Model

The superiorities of servo motors include high precision and rapid response. In the case of the AC servo system, the driving force output by PMLSM is obtained through the feedback gains of position, speed, and current after proper simplification of the reference signal. The current equation is formulated as follows:

$$L\dot{i} + R'i = K_{ip}\left[K_{vp}\left(K_{pp}K_v e - v\right) - i\right] - K_{emf}v \tag{23}$$

where $L$ and $R'$ separately signify the armature inductance and resistance of the motor; $i$ denotes the current of the motor; $e$ stands for the displacement error ($e = x_r - x_t$, $x_r$ the ideal input); $K_{ip}$, $K_{vp}$ and $K_{pp}$ refer separately to the current, speed, and position loop gains; $K_v$ denotes the command velocity regulation gain; $v$ represents the output velocity of the table; and $K_{emf}$ represents the motor back-EMF coefficient.

The voltage input to the servo motor $V_m$ after overcoming the back EMF $V_E$ leads to a time-delayed motor current $i$, resulting in multiplication of the motor driving force constant $K_M$. Accordingly, the linear motor's output driving force is derivable by

$$F_s = K_M i \tag{24}$$

### 3.4. Dual Linear-Motor Differential-Drive System Block Diagram Model

For our dual linear-motor differential-drive servo system, its mechanical formula is converted into a block diagram for the transfer function, as depicted in Figure 5. A holistic model of electromechanical coupling dynamics is identified for our system, where the influence of nonlinear friction is taken into account.

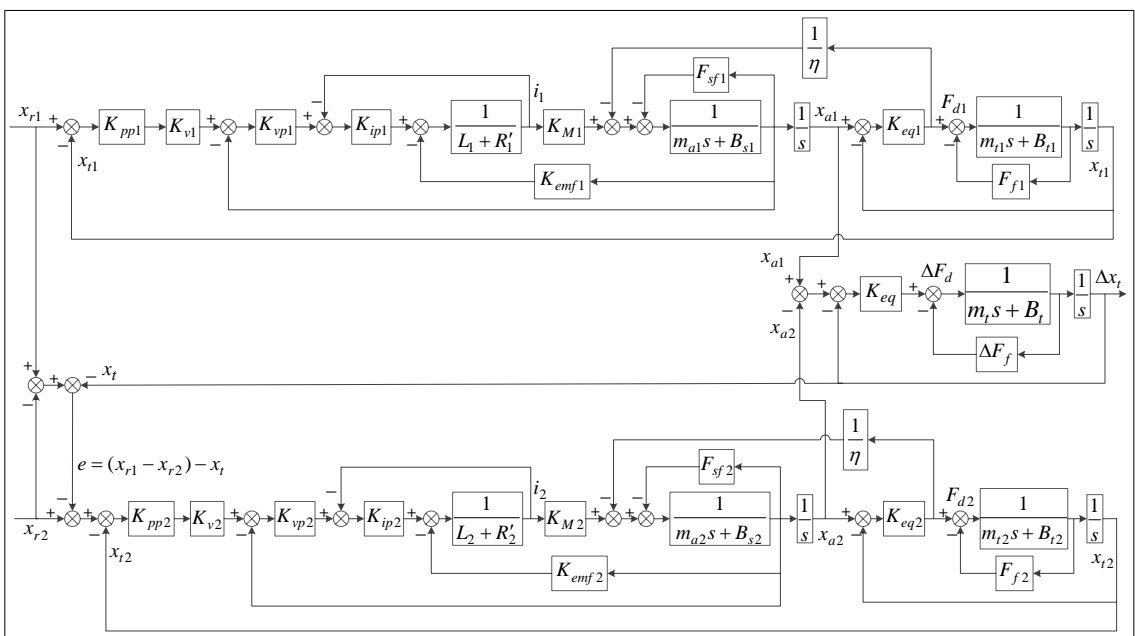

**Figure 5.** Transfer function block diagram of the dual linear-motor differential-drive system.

## 4. Simulation and Analysis

The model parameters are either sourced from manufacturers' catalogs, approximated from prior information, or computed from the drive element CAD models, as detailed in Table 1.

**Table 1.** Parameters and related values used in modeling the dual linear-motor differential-drive system.

| Parameters | Given Value |
| --- | --- |
| position loop gain $K_{pp}$ | 7.5 |
| command velocity adjusting the gain $K_v/\text{m} \cdot (\text{V} \cdot \text{s})^{-1}$ | 150 |
| velocity loop gain $K_{vp}/\text{A} \cdot \text{s} \cdot \text{m}^{-1}$ | 25 |
| current loop gain $K_{ip}/\text{V} \cdot \text{A}^{-1}$ | 5 |
| motor driving force constant $K_M/\text{N} \cdot \text{A}^{-1}$ | 0.75 |
| back EMF coefficient $K_{emf}/\text{V} \cdot \text{s} \cdot \text{m}^{-1}$ | 0.2 |
| motor armature inductance $L/\text{mH}$ | 5.5 |
| motor armature resistance $R'/\Omega$ | 1 |
| the equivalent mass of the linear motor actuator $m_a/\text{Kg}$ | 12 |
| the equivalent viscous friction coefficient on the linear motor actuator $B_s/\text{N} \cdot \text{s} \cdot \text{m}^{-1}$ | 2 |
| equivalent mass of the table $m_t/\text{Kg}$ | 50 |
| guide viscous friction coefficient $B_t/\text{N} \cdot \text{s} \cdot \text{m}^{-1}$ | 8 |
| the comprehensive equivalent stiffness $K_{eq}/\text{N} \cdot \text{m}^{-1}$ | $2.06 \times 10^7$ |
| static friction $f_s/\text{N}$ | 25 |
| coulomb friction $f_c/\text{N}$ | 15 |
| Stribeck velocity $v_s/\text{m} \cdot \text{s}^{-1}$ | 0.0012 |
| transmission efficiency $\eta$ | 0.9 |

### 4.1. Critical Creeping Speed Analysis

Through repeated computation of system parameters, the critical creeping velocity of the table is derived for a linear-motor single-drive system, whose value is about 2.0 mm/s. Figures 6–9 depict the simulations of this system when the feed rates are separately 1.9-, 2.0-, 2.1-, and 2.5 mm/s. The critical creeping velocity is considered to be attained when the table state is steady and the feed rate does not fluctuate pronouncedly.

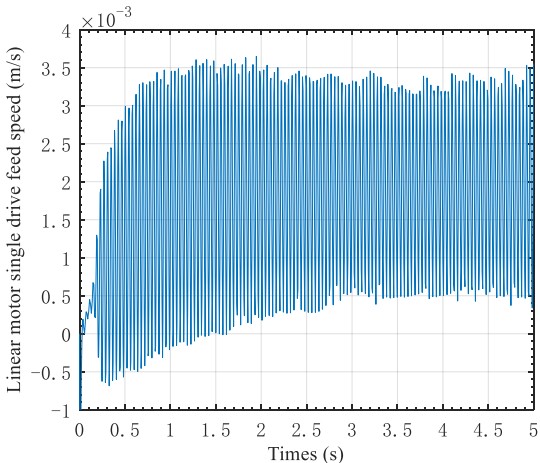

**Figure 6.** Linear-motor single-drive feed speed (1.9 mm/s).

To identify the lowest steady-state feed velocity of a linear-motor single-drive system, four magnitudes (1.9-, 2.0-, 2.1-, and 2.5 mm/s) are assessed by simulation. As displayed in Figure 6, evident vibration along with limit cycle phenomena are present in the output velocity curve for the linear-motor single-drive system at a table feed velocity of 1.9 mm/s. Nonetheless, as illustrated in Figure 7, the output velocity curve attains a steady state at a table feed velocity of 2.0 mm/s. According to Figures 8 and 9, the output velocity

curve attains the steady state following approximately 2 s of adjustment when the table feed velocities are severally 2.1- and 2.5 mm/s. After overall analysis, the linear-motor single-drive system is considered to have a critical creeping velocity of 2.0 mm/s.

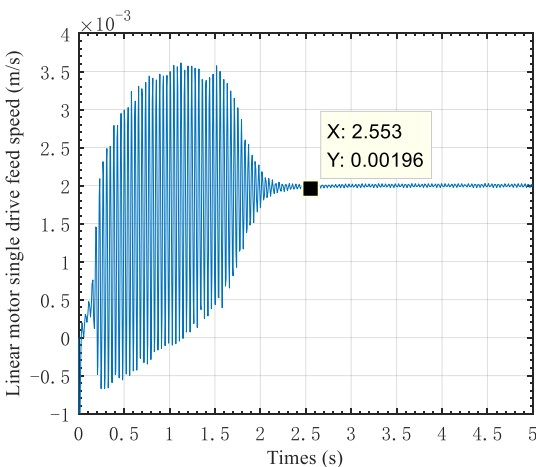

**Figure 7.** Linear-motor single-drive feed speed (2.0 mm/s).

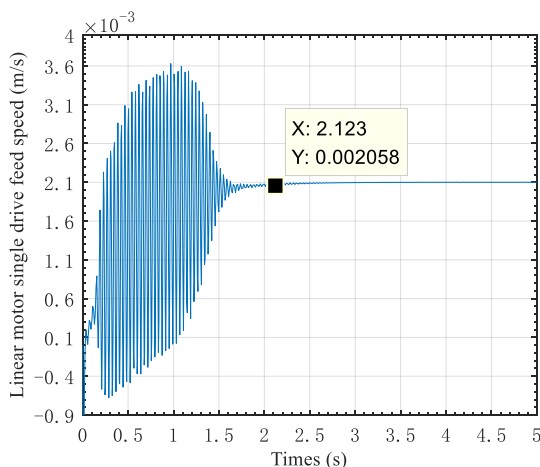

**Figure 8.** Linear-motor single-drive feed speed (2.1 mm/s).

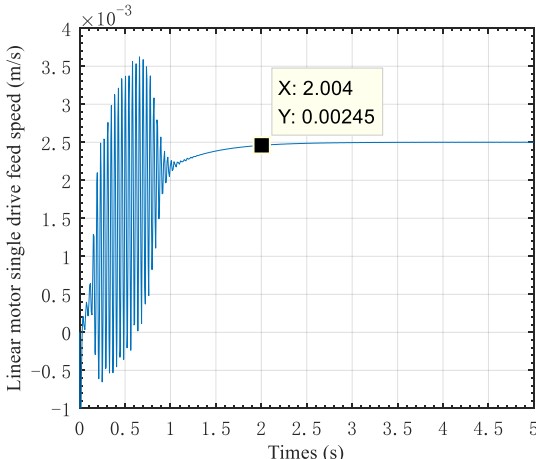

**Figure 9.** Linear-motor single-drive feed speed (2.5 mm/s).

In the case of a dual linear-motor differential-drive system, to identify its critical creeping velocity, the constant rate is introduced into the under and upper linear motors, respectively, whose value considerably exceeds (20 times at maximum) that derived with

the single-drive system. A few simulations were conducted under identical parameters, and the dual linear-motor differential-drive system attains a critical creeping speed of about 1.0 mm/s.

To find the critical creeping speed for the dual linear-motor differential-drive system, we input two constant rates separately to the upper and under linear motors via the control system, thereby guaranteeing the upper and under table instantaneous approximate synchronization. In this paper, the upper linear motor rate is set to 40.9-, 41-, 41.1-, and 41.5 mm/s, respectively, while the under linear motor rate is set to 40 mm/s. Simulation results for the dual linear-motor differential-drive system under 0.9-, 1.0-, 1.1-, and 1.5 mm/s feed rates are shown, respectively. The results obtained are shown in Figures 10–13.

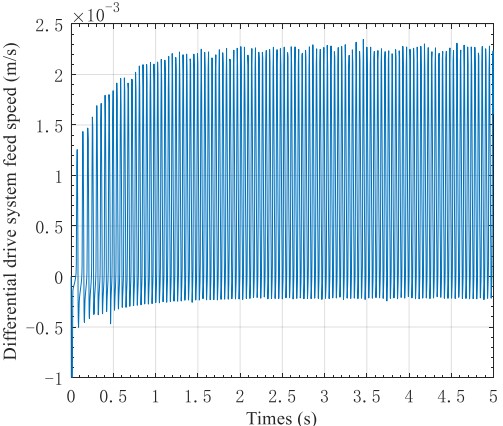

**Figure 10.** Differential-drive system feed speed (0.9 mm/s).

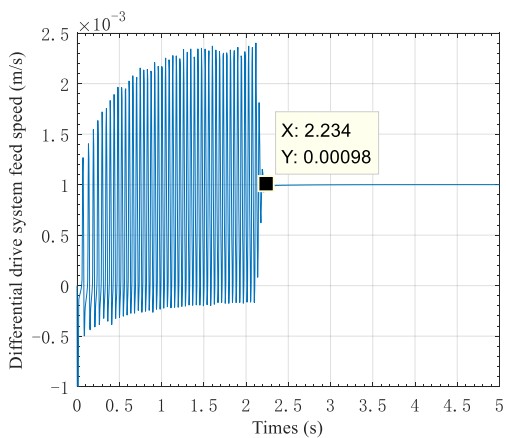

**Figure 11.** Differential-drive system feed speed (1.0 mm/s).

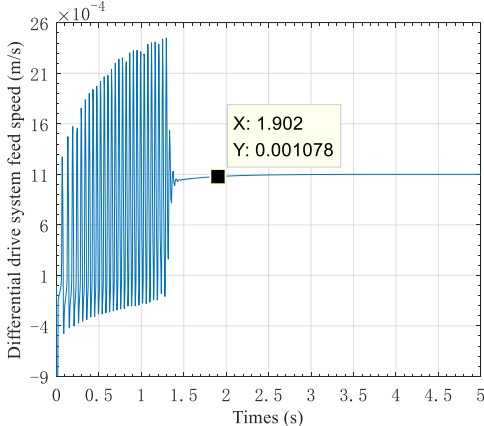

**Figure 12.** Differential-drive system feed speed (1.1 mm/s).

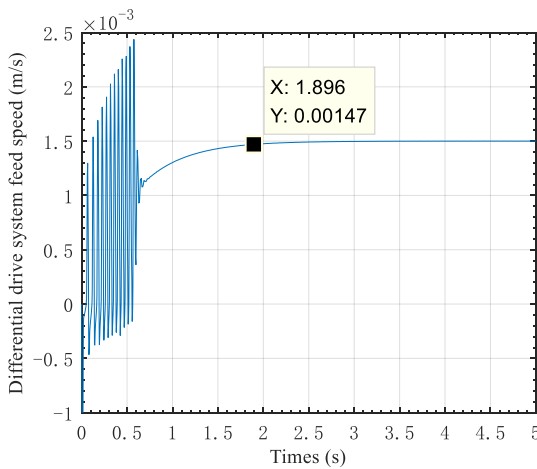

**Figure 13.** Differential-drive system feed speed (1.5 mm/s).

It is clear from Figure 10 that when the table feed velocity is below the critical creeping velocity, the system feed characteristics exhibit oscillation along with the limit-cycle phenomenon. Nonetheless, as displayed in Figure 11, when the table feed velocity exceeds the critical creeping velocity, the system is capable of attaining a steady state after a certain duration of adjustment (approximately 2.234 s at a 1.0 mm/s feed velocity), and our dual linear-motor differential-drive system boasts excellent feed characteristics at low velocity.

According to Figure 10, when the dual linear-motor differential-drive system has a 0.9 mm/s feed velocity, evident vibration along with limit-cycle phenomena are present in the output velocity curve for the system. Nonetheless, as is clear from Figures 11–13, the system output velocity curve attains the steady state following adjustment severally at 1.0-, 1.1-, and 1.5 mm/s feed velocities. The dual linear-motor differential-drive system is considered to have a critical creeping velocity of 1.0 mm/s. Through repeated computation of system parameters, the foregoing four magnitudes (0.9, 1.0, 1.1, and 1.5 mm/s) are identified to mostly approximate the critical creeping velocity. Hence, critical creeping velocity for our dual linear-motor differential-drive system is explored by choosing these four magnitudes.

As is clear from Figures 6–13, our dual linear-motor differential-drive system exhibits lower critical creeping velocity than the linear-motor single-drive system under identical parameters. As shown in Figure 14, the upper and under linear motor rates are set separately to 42- and 40 mm/s. The output result of table velocity is further studied when the constant feed velocity for the dual linear-motor differential-drive system is the critical creeping velocity for the linear-motor single-drive system as 2.0 mm/s.

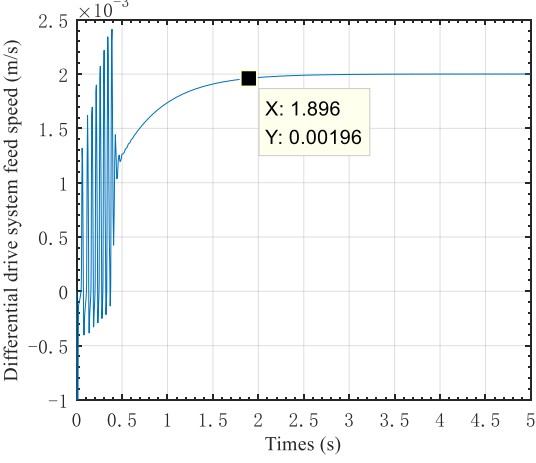

**Figure 14.** Differential-drive system feed speed (2.0 mm/s).

As is displayed in Figures 6–14, the critical creeping velocity of our dual linear-motor differential-drive system is lower, which can provide the table with a comparatively normal and stable working condition upon reaching the linear-motor single-drive system to its critical state. In addition, our system exhibits superior dynamic response characteristics to the linear-motor single-drive system, which thus outperforms it in terms of low-velocity stability.

Comparing Figures 7 and 11, it can be seen that due to the influence of frictional nonlinearity, the system undergoes multiple oscillations before reaching a steady state. when the feed rates of the linear-motor single-drive system and the dual linear-motor differential-drive system are both 2 mm/s, the adjustment time for the linear-motor single-drive system to reach steady state (error = 2%) is 2.553 s, while the adjustment time for the dual linear-motor differential-drive system to reach steady state (error = 2%) is 1.896 s. From a quantitative analysis perspective, the response speed of the dual linear-motor differential-drive system is 25.73% higher than that of the linear-motor single-drive system. At the same time, comparative analysis shows that the oscillation frequency of the dual linear-motor differential-drive system is 23.1% less than that of the linear-motor single-drive system.

### 4.2. Constant Speed Analysis

The analysis reveals that our dual linear-motor differential-drive system attains a critical creeping velocity of about 1.0 mm/s. The discussion here emphasizes the identification of output variation in case the velocities of both linear motors exceed the table critical creeping velocity for the linear-motor single-drive system, yielding a 1.0 mm/s resultant rate. The settings for the upper linear-motor rate are 3-, 4-, 5- and 6 mm/s, separately, while the corresponding command speeds of the under linear motor are 2-, 3-, 4-, and 5 mm/s, the simulation results are shown in Figures 15–18, respectively.

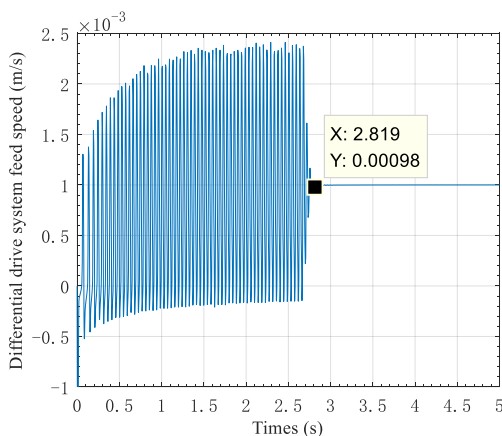

**Figure 15.** The two motor speeds are 3 mm/s and 2 mm/s.

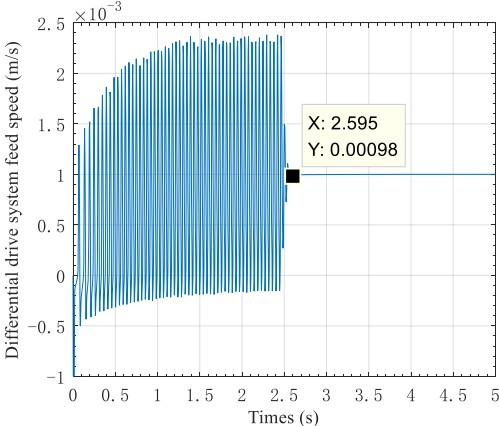

**Figure 16.** The two motor speeds are 4 mm/s and 3 mm/s.

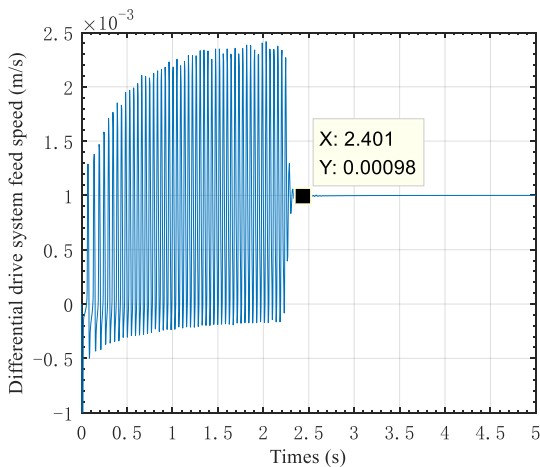

**Figure 17.** The two motor speeds are 5 mm/s and 4 mm/s.

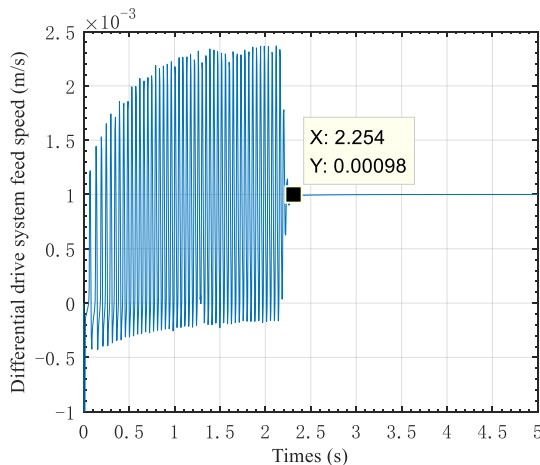

**Figure 18.** The two motor speeds are 6 mm/s and 5 mm/s.

From Figures 11 and 15, Figures 16–18, it can be seen that when the minimum command speed of the upper linear motor and the under linear motor are both greater than 5 mm/s, and the differential value exceeds the critical creeping velocity of our dual linear-motor differential-drive system by 1.0 mm/s. The adjustment time of the output speed of the dual linear-motor differential-drive system is almost unchanged. Therefore, to ensure the stable output speed of the dual linear-motor differential-drive system, it is required that the minimum command speed of the upper linear motor and under linear motor are not less than 5 mm/s.

To further investigate the minimum speed difference between the upper linear motor and under linear motor, the output speed of the dual linear-motor differential-drive system will not produce over modulation. Set the command speed of the upper linear motor to 7.3 mm/s and the command speed of the under linear motor to 5 mm/s, the simulation result is shown in Figure 19. Similarly, set the command speed of the upper linear motor to 42.3 mm/s and the command speed of the under linear motor to 40 mm/s, the simulation result is shown in Figure 20. From Figures 19 and 20, it can be concluded that when the command speeds of the upper linear motor and under linear motor are both higher than 5 mm/s and the speed difference is not less than 3 mm/s, the output speed of the dual linear-motor differential-drive system will not produce over modulation.

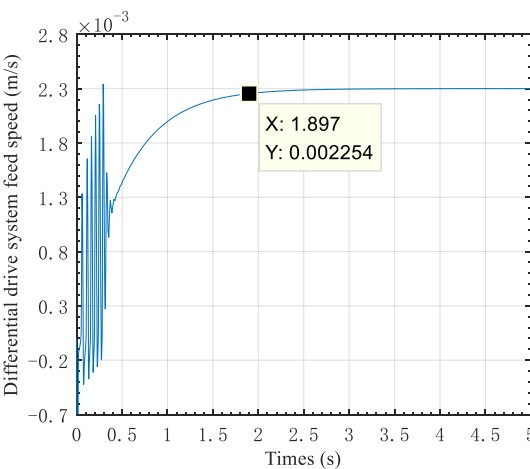

**Figure 19.** The two motor speeds are 7.3 mm/s and 5 mm/s.

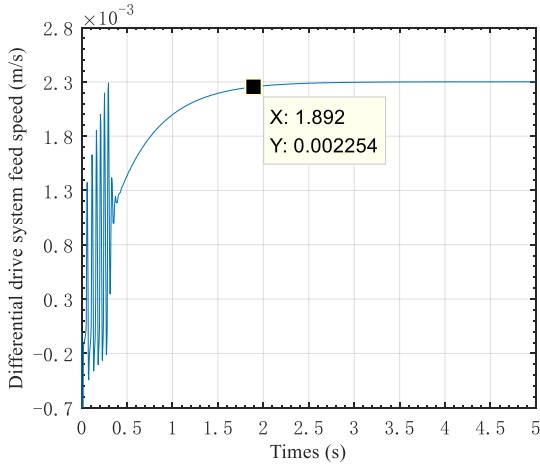

**Figure 20.** The two motor speeds are 42.3 mm/s and 40 mm/s.

The authors also compare the differences in rate fluctuations between the dual linear-motor differential- and linear-motor single-drive systems when the former gives a feed rate exceeding its own critical creeping velocity to the latter. The feed velocity of the latter is 2.3 mm/s, and Figure 21 depicts the result. As shown in Figure 22, the linear-motor single-drive system does not experience over-modulation when the feed speed is 3.7 mm/s. The rate of the upper linear motor is set to 43.7 mm/s, the corresponding command speed of the under linear motor to 40 mm/s, and Figure 23 presents the simulation outcome. Comparing Figures 20 and 21 and Figures 22 and 23, the fluctuation of the dual linear-motor differential-drive system is smaller than the linear-motor single-drive system when the rate settings are low for both. In addition, the former consumes less adjustment time, attains quicker response, and superior low-velocity micro-feed performance.

From Figures 20 and 21, it can be concluded that before the system reaches a steady state, the oscillation frequency of the dual linear-motor differential-drive system is 77.78% less than that of the linear-motor single-drive system. In addition, the response speed of the dual linear-motor differential-drive system is 6.89% higher than that of the linear-motor single-drive system.

From Figures 22 and 23, it can be concluded that before the system reaches a steady state, the oscillation frequency of the dual linear-motor differential-drive system is 60% less than that of the linear-motor single-drive system. At the same time, the response speed of the dual linear-motor differential-drive system is 3.13% higher than that of the linear-motor single-drive system.

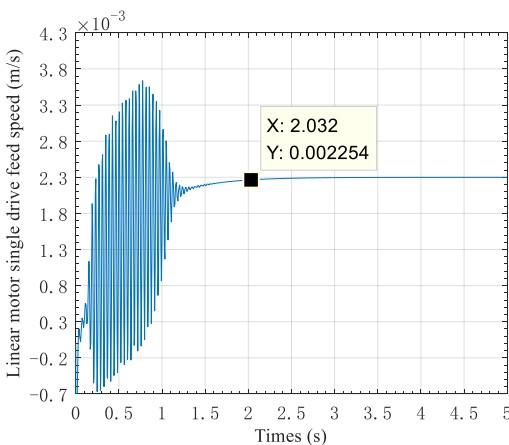

**Figure 21.** Linear-motor single-drive feed speed (2.3 mm/s).

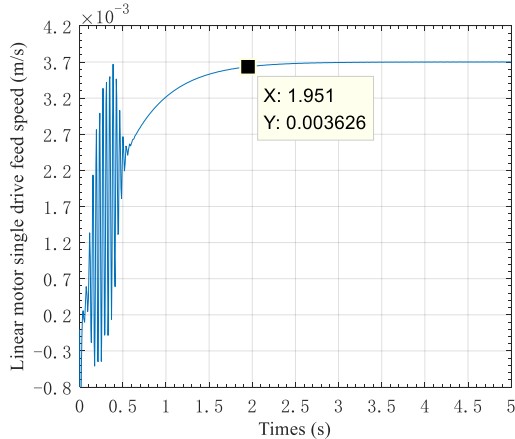

**Figure 22.** Linear-motor single-drive feed speed (3.7 mm/s).

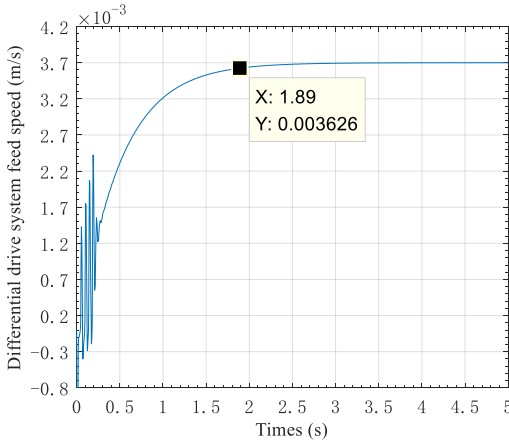

**Figure 23.** The two motor speeds are 43.7 mm/s and 40 mm/s.

### 4.3. Variable Speed Analysis

According to the foregoing constant speed analysis, the linear-motor single-drive system has a critical creeping velocity of 2 mm/s, and its feed speed without over modulation is 3.7 mm/s. The dual linear-motor differential-drive system does not produce over-modulation and has a feed rate of 2.3 mm/s. Further, the output velocity difference between the linear-motor single-drive system and the dual linear-motor differential-drive system is explored under variable velocity conditions.

In the case of the linear-motor single-drive system, sinusoidal velocity signal with amplitude of 2 mm/s, the result is shown in Figure 24. For the dual linear-motor differential-

drive system, the sinusoidal velocity signal is set to 42 mm/s for the upper linear motor, and to 40 mm/s for the under linear motor, the result is shown in Figure 25. Similarly, in the case of the linear-motor single-drive system, sinusoidal velocity signal with amplitude of 2.3 mm/s, the result is shown in Figure 26. For the dual linear-motor differential-drive system, the sinusoidal velocity signal is set to 42.3 mm/s for the upper linear motor, and to 40 mm/s for the under linear motor, the result is shown in Figure 27. In the case of the linear-motor single-drive system, sinusoidal velocity signal with amplitude of 3.7 mm/s, the result is shown in Figure 28. For the dual linear-motor differential-drive system, the sinusoidal velocity signal is set to 43.7 mm/s for the upper linear motor, and to 40 mm/s for the under linear motor, the result is shown in Figure 29.

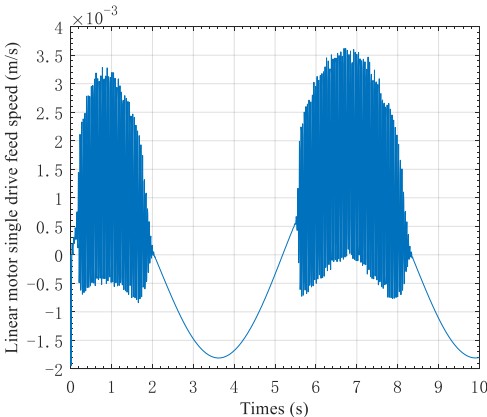

**Figure 24.** Single-drive sinusoidal feed (Am 2.0 mm/s).

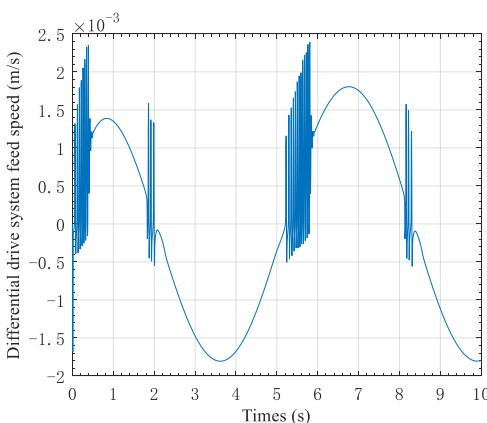

**Figure 25.** Differential-drive sinusoidal feed (Am 2.0 mm/s).

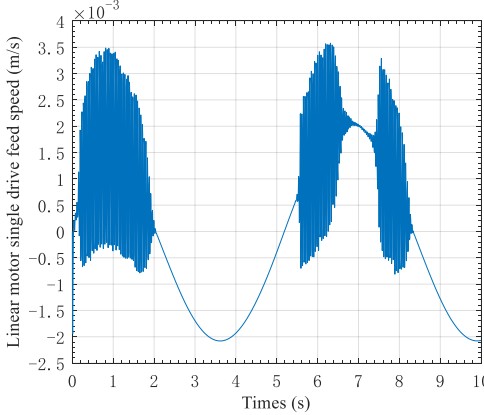

**Figure 26.** Single-drive sinusoidal feed (Am 2.3 mm/s).

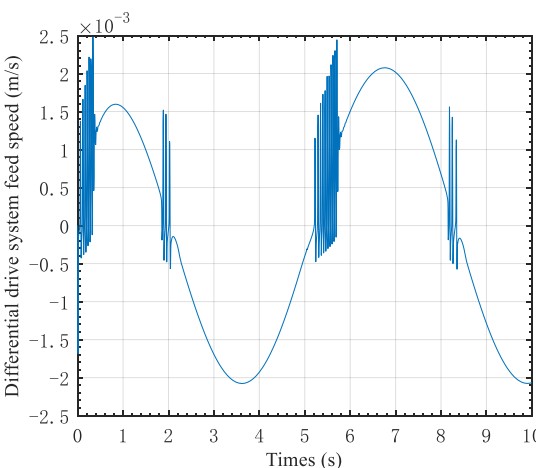

**Figure 27.** Differential-drive sinusoidal feed (Am 2.3 mm/s).

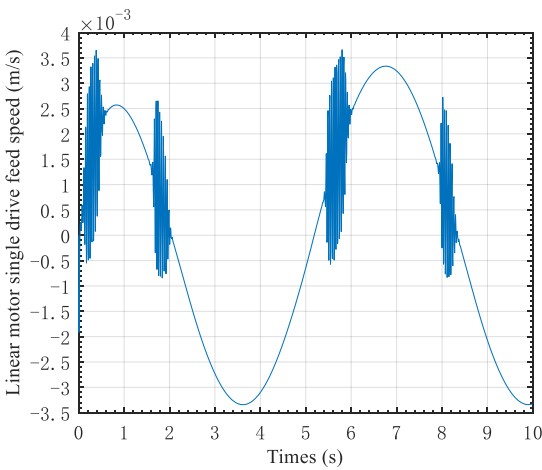

**Figure 28.** Single-drive sinusoidal feed (Am 3.7 mm/s).

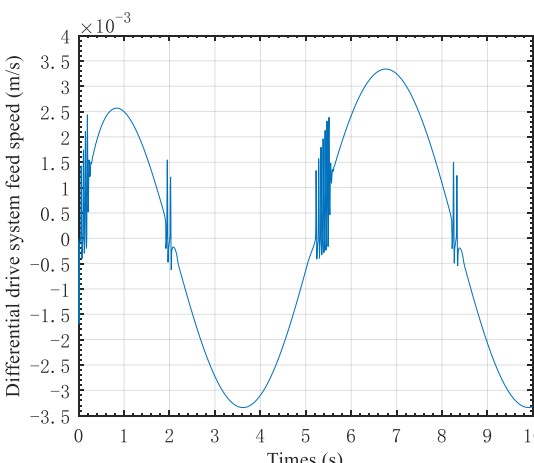

**Figure 29.** Differential-drive sinusoidal feed (Am 3.7 mm/s).

Comparing Figures 24 and 25, Figures 26 and 27, and Figures 28 and 29, the fluctuation of the dual linear-motor differential-drive system is found to be smaller than the linear-motor single-drive system when the sinusoidal velocities are low for both, indicating that the former outperforms the latter regarding variable-speed micro feed. As is also clear from Figures 24–29, when the velocity crosses zero point, evident velocity fluctuation takes place

in both the dual linear-motor differential- and linear-motor single-drive systems, with the former exhibiting markedly weaker fluctuation than the latter.

Comparing Figures 24 and 25, Figures 26 and 27, and Figures 28 and 29, it can be concluded that the maximum error in the output speed of the dual linear-motor differential-drive system is reduced by 75%, 84.62%, and 25%, respectively, compared to the maximum error in the output speed of the linear-motor single-drive system. Meanwhile, when both the dual linear-motor differential-drive system and the linear-motor single-drive system do not produce over modulation, the maximum speed error of the two systems is not significantly affected by the input alternating speed.

## 5. Conclusions

With the proposed dual linear-motor differential-drive micro-feed mechanism, the shortcomings of the existing CNC machine tool unit axis can be overcome, which can hardly achieve precise and uniform micro-feed motion due to the impact of nonlinear creeping. In addition, in contrast to the currently popular micro-feed mechanisms exploiting the piezoelectric, thermoelastic, and electric (magnetic) effect techniques, our dual linear-motor differential-drive system also has superiorities like large stroke, convenient control, as well as high rigidity, load, and precision. Thus, this research obviously has a profound significance to the new concept design of CNC equipment, the improvement of processing performance, as well as the development of ultra-precision machining technology.

(1) A dual linear-motor differential-drive system has been designed and the dynamic model for electromechanical coupling of the system has been created by a lumped parameter approach. In addition, the transfer function block diagram has also been plotted for simulating such system models as mechanical, motor, and frictional, where the closed-loop feeding system is taken into account.

(2) Numerical simulations reveal that the critical creeping velocity of a dual linear-motor differential-drive micro-feed system is lower than that of a linear-motor single-drive feed system. For a particular dual linear-motor differential-drive micro-feed system, the output velocity is impacted by the integration of two different linear motor feed velocities. By numerical analysis, the differences between the lowest stable feed velocity from the minimum feed velocity of our dual linear-motor differential-drive system can be obtained.

(3) As found by the output velocity analysis under the fixed and variable velocity conditions for the dual linear-motor differential-drive micro-feed system and the linear-motor single-drive feed system, the former boasts faster responsiveness and superior low-velocity micro-feed performance.

(4) The results of theoretical computation and numerical analysis fundamentally agree with the actual engineering phenomenon, suggesting the rationality of the created models. The establishment of the system model paves the ground for further research concerning controller design.

## 6. Patents

Yu H W, Geng F Q, Wang C, et al. A Dual Linear-Motor Differential Micro-Feed Servo System and Control Method [P]. Chinese invention patent, 2021, ZL 2020 1 0517725.3 (In Chinese) [26].

**Author Contributions:** H.Y. was responsible for the conception and design, acquisition of data, analysis and interpretation of data, drafting the initial manuscript, and revising it critically for important intellectual content. G.Z. collected the data and pictures, completed the follow-up information, and wrote the draft. Y.L. was responsible for the conception and design, interpretation of data, and reviewing all drafts of the manuscript. J.Z., G.W. and H.J. were responsible for the numerical analysis of the manuscript and provided valuable suggestions for the manuscript. All authors have read and agreed to the published version of the manuscript.

**Funding:** This research was funded by the Doctoral Research Fund Project of Shandong Jianzhu University (Grant No. X21030Z).

**Institutional Review Board Statement:** Not applicable.

**Informed Consent Statement:** Not applicable.

**Data Availability Statement:** The original contributions presented in the study are included in the article, further inquiries can be directed to the corresponding author.

**Conflicts of Interest:** The authors declare no conflicts of interest.

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
