# Peer review of "Research on the Dynamic Characteristics of a Dual Linear-Motor Differential-Drive Micro-Feed Servo System"

_applsci, doi:10.3390/app14083170_

Round 1

Reviewer 1 Report

Comments and Suggestions for Authors

Dear Authors,

The topics of the work are related to machine drive systems. The authors presented a numerical study of the differential drive micro feed servo system. I have a few comments on the work presented.

Major revision:

1.     The model presented in the paper and the paper itself is very close to: Yu, H.W.; Zhang, Z.Z.; Xing, J.F. Micro feed characteristic analysis of a new crawler guide rail dual drive servo system. Science 587 Progress 2021, 104, 1-24,

2.     Why was a series connection of springs rather than a parallel connection used in the mechanical model of linear motor single drive system. As shown in Fig. 2 and Fig. 3. The servo motor system with actuator and carriage is connected in parallel.

3.     Fig. 4 shows the drawing already presented in the paper: Yu, H.W.; Zhang, Z.Z.; Xing, J.F. Micro feed characteristic analysis of a new crawler guide rail dual drive servo system. Science 587 Progress 2021, 104, 1-24, cite it.

4.     What do ranges I, II, III, IV mean in Fig. 4.

5.     Fig. 5 presents a block diagram of the drive system.  How the kinematic excitation signals x_r_1 and x_r_2 are generated.

6.     On what basis were the parameters in Table 1 assumed. 

7.     Fig. 24 shows the “Single drive sinusoidal feed (Am 2.0mm/s)”. What is the reason that the harmonic waveform is not symmetrical. 

Reviewer

Reviewer 2 Report

Comments and Suggestions for Authors

The article discusses the differential drive system, which was implemented by using two linear motors. This solution is intended to eliminate control inaccuracies caused by friction nonlinearity occurring during movement at low speeds.

The authors conduct a literature review and discuss the methods used to solve this problem so far.

In the article, the authors provide a mechanical, kinematic and dynamic model of proposed solution. A friction model is also proposed, which is of key importance in the analysis of the mechanism under consideration.

Later in the article, the authors carry out simulation calculations for sample data. The results of calculations of selected parameters are given, which are then analyzed. The article ends with conclusions.

The layout of the article and its content meet the conditions of a scientific article.

Additionally, the method proposed by the authors has been submitted for a patent, which proves the novelty of the topic described - therefore it can be expected that the article will be of interest to the recipients.

Technical Note:

Fig. 1 indicates the individual components of the system with numbers. The names of the indicated parts are given in the text. These names should also be provided below the drawing.

Reviewer 3 Report

Comments and Suggestions for Authors

This article presents, as the authors write, a dual linear motor differential drive micro feed servo system, mainly through the optimization design of the transmission mechanism. Owing to the differential synthesis of micro feed from the upper and under linear motors, the impact of friction nonlinearity during ultra-low velocity micro feed is avoided, endowing the system with a lower stable feed speed to achieve precise micro feed control. Through simulation, the differences in response between the linear motor single drive system and the dual linear motor differential drive system are examined under fixed or variable feeding velocity, as well as the impact of varying velocity combinations of dual linear motors on the output speed of differential drive system.

The authors presented the results of a qualitative analysis of differences between the linear motor single drive system and the dual linear motor differential drive system are examined under fixed or variable feeding velocity, as well as the impact of varying velocity combinations of dual linear motors on the output speed of differential drive system.

As the authors write, many numerical simulations were performed, but unfortunately no qualitative results of these analyzes were presented. A verbal discussion of the sample results shown in the figures cannot be called an analysis of the results obtained. It was necessary to present quantitative analysis results for the obtained results of numerical simulations. Therefore, the article can be considered as a contribution to more serious, future research and analyses. The work requires supplementation with the results of quantitative analyses, among others, of the impact of the friction phenomenon, which is non-linear in reality, on the speed of the mechanism in the area of ​​small values.

Round 2

Reviewer 1 Report

Comments and Suggestions for Authors

Thank you for your detailed explanation.
Good work.
Reviewer

Reviewer 3 Report

Comments and Suggestions for Authors

After introducing explanations and additions, I have no comments.